# Evaluation of the Chemotherapy Drug Response Using Organotypic Cultures of Osteosarcoma Tumours from Mice Models and Canine Patients

**DOI:** 10.3390/cancers13194890

**Published:** 2021-09-29

**Authors:** Bénédicte Brulin, John C. Nolan, Tecla Marangon, Milan Kovacevic, Mathias Chatelais, Pierre Meheust, Jérome Abadie, Louis-Romée Le Nail, Philippe Rosset, Meadhbh Á. Brennan, Pierre Layrolle

**Affiliations:** 1INSERM, UMR 1238, PHY-OS, Bone Sarcomas and Remodelling of Calcified Tissues, School of Medicine, University of Nantes, 1 rue Gaston Veil, 44035 Nantes, France; benedicte.brulin@inserm.fr (B.B.); mathias.chatelais@profile-hit.com (M.C.); lr.lenail@chu-tours.fr (L.-R.L.N.); philippe.rosset@chu-tours.fr (P.R.); 2INSERM, UMR 1214, ToNIC, Université Paul Sabatier, CHU Purpan, Pavillon Baudot, Place du Dr Baylac, 31024 Toulouse, France; 3Biomedical Engineering, School of Engineering, and the Regenerative Medicine Institute (REMEDI), School of Medicine, National University of Ireland Galway (NUIG), H91 TK33 Galway, Ireland; jnolan@nuigalway.ie (J.C.N.); T.Marangon1@nuigalway.ie (T.M.); M.KOVACEVIC1@nuigalway.ie (M.K.); meadhbh.brennan@nuigalway.ie (M.Á.B.); 4Vetoceane, 9 Allée Alphonse Fillion, 44120 Vertou, France; drpierremeheust@gmail.com; 5ONIRIS, Veterinary School of Nantes, 101 Route de Gachet, 44300 Nantes, France; jerome.abadie@oniris-nantes.fr; 6CHRU Tours, Service de Chirurgie Orthopédique et Traumatologique, Hôpital Trousseau, Université François-Rabelais de Tours, 37044 Tours, France

**Keywords:** osteosarcoma, biopsy, organotypic culture, chemotherapy testing, tumour microenvironment

## Abstract

**Simple Summary:**

Osteosarcoma is a bone cancer with 75% of cases occurring in people younger than 25 years old. 35–45% of patients demonstrate resistance to chemotherapeutics and critically, survival rates for osteosarcoma is only 10–30% for patients with metastases. Therefore, reliable and patient-specific drug testing modalities are needed. Organotypic slice culture consists of sections of tumours, which survive and preserve the tumours mechanical and cellular properties, thereby enabling personalised testing of drugs. This study aimed to characterise organotypic slice cultures of osteosarcoma bone tumours derived from mice and dogs and to use these models for testing of anti-tumoural drugs. This study reports the various cell constituents of the model and the maintenance of osteosarcoma organotypic cultures over several weeks. A significantly decreased sensitivity to chemotherapy in 3D organotypic culture relative to 2D monolayer was found, highlighting the need to test anti-cancer drugs in a more personalized and biomimetic manner.

**Abstract:**

Improvements in the clinical outcome of osteosarcoma have plateaued in recent decades with poor translation between preclinical testing and clinical efficacy. Organotypic cultures retain key features of patient tumours, such as a myriad of cell types organized within an extracellular matrix, thereby presenting a more realistic and personalised screening of chemotherapeutic agents ex vivo. To test this concept for the first time in osteosarcoma, murine and canine osteosarcoma organotypic models were maintained for up to 21 days and in-depth analysis identified proportions of immune and stromal cells present at levels comparable to that reported in vivo in the literature. Cytotoxicity testing of a range of chemotherapeutic drugs (mafosfamide, cisplatin, methotrexate, etoposide, and doxorubicin) on murine organotypic culture ex vivo found limited response to treatment, with immune and stromal cells demonstrating enhanced survival over the global tumour cell population. Furthermore, significantly decreased sensitivity to a range of chemotherapeutics in 3D organotypic culture relative to 2D monolayer was observed, with subsequent investigation confirming reduced sensitivity in 3D than in 2D, even at equivalent levels of drug uptake. Finally, as proof of concept for the application of this model to personalised drug screening, chemotherapy testing with doxorubicin was performed on biopsies obtained from canine osteosarcoma patients. Together, this study highlights the importance of recapitulating the 3D tumour multicellular microenvironment to better predict drug response and provides evidence for the utility and possibilities of organotypic culture for enhanced preclinical selection and evaluation of chemotherapeutics targeting osteosarcoma.

## 1. Introduction

Osteosarcoma is a primary solid malignancy of the bone with an annual global incidence rate of 3–5 cases per million [1]. It usually manifests in the long bones, such as femur, tibia, and humerus, but, more rarely, involves the skull and the pelvic area. Osteosarcoma affects children and adolescents in particular due to the pubertal growth spurt, during which bone growth occurs at a faster rate [1]. Fifteen percent of osteosarcoma patients develop metastases, most commonly in the lungs [1], which reduce the 5-year survival rate from 70% to 30% [2]. Recurrence, whether local or distant, occurs in 30–40% of patients; ~30% of patients survive when recurrence is local, however, if they develop metastases, survival rates decrease to ~10% [3]. Current therapy for osteosarcoma includes a cycle of neoadjuvant chemotherapy, followed by surgical tumour resection, and an additional chemotherapy cycle [2]. Osteosarcoma chemotherapy includes ifosfamide and cisplatin (alkylating agents), methotrexate (nucleotide synthesis inhibitor), and etoposide and doxorubicin (DNA re-ligation inhibitors). In most countries, the standard osteosarcoma regime comprises MAP (high-dose methotrexate, doxorubicin, and cisplatin) [4]. In France, the standard preoperative chemotherapy combines methotrexate, etoposide and ifosfamide (M-EI) based on the OS94 trial [5]. Unfortunately, survival rates for osteosarcoma have not improved over the past few decades with chemoresistance recognised as a primary contributing factor [6], and 35–45% of patients developing resistance to antitumoural drugs [3].

The stroma of the tumour is composed of the extracellular matrix, bone cells, endothelial cells, immune cells, and mesenchymal stem cells (MSCs), both undifferentiated and differentiated into cancer-associated fibroblasts (CAFs) [7,8]. Despite the tumour-suppressing abilities of some populations, the stroma enhances growth, invasion, and metastatic abilities after malignant transformation [9]. Stromal MSCs secrete lactate due to oxidative stress caused by the tumour, which is then taken up by osteosarcoma cells, enhancing their mitochondrial activity [10]. CAFs act as pro-tumorigenic agents, creating a microenvironment in which tumour proliferation is promoted, and protecting it against chemotherapeutics [11].

Macrophages (M0/M1/M2) represent over half of the immune population (circa 60%) in osteosarcoma tumours [12]. Macrophages present in the tumour microenvironment (TME) are referred to as tumour-associated macrophages (TAMs); they resemble M2 polarised cells, display anti-inflammatory properties, and play a pivotal role in modulating tumour survival [13,14]. The role of TAMs in osteosarcoma is uncertain; they have been shown to promote tumour growth and spread, as well as suppressing anti-tumour immune responses [15,16,17]; however, they have also been associated with reduced metastasis and improved survival [18,19].

Typically, drug screening is performed in 2D monolayer cell lines, which fail to consider the complex 3D environment in vivo or the myriad of cell types that comprise the tumour. To better mimic these features in vitro, 3D tumour models have been developed. Studies comparing the efficacy of drugs in in vitro 3D models, such as spheroid cultures or cells seeded onto 3D scaffolds, to efficacy in traditional 2D cultures show that cells grown in 3D are less sensitive to the effect of chemotherapeutic drugs than their monolayer counterpart [20,21,22,23]. While substantial advances have been made in the field of biomimetic tissue-engineered constructs, these 3D in vitro models, which comprise typically one or two cell types, still fail to recapitulate the ECM microenvironment, and the cellular complexity of the native tumour.

Organotypic culture consists of slices of tissue (100–400 μm thickness) taken from an organ or tumour, which survive and preserve the tissue’s properties. Typically, organotypic slices can be cultured for a timeframe from weeks to months; hence, they are useful in vitro models for biological studies [24]. Of particular interest is the use of organotypic cultures to study the effects of drugs in vitro in a context that better simulates the in vivo environment. While chemotherapy drugs have been screened using various organotypic models of cancers in organs and tissues [25,26], few studies have explored organotypic models of bone tumours [27]. Given the immense promise of this model as a biomimetic, cost-efficient, and non-labour-intensive platform for reliable and personalised chemotherapy screening, the development of an organotypic model for osteosarcoma is the focus of the current work.

Briefly, in this study an organotypic mouse model of osteosarcoma was used to study the efficacy of chemotherapeutics, and, as proof-of-concept, organotypic cultures from canine patients were also investigated. These models demonstrate organotypic culture of osteosarcoma over weeks in vitro and detail the various cell constituents, particularly stromal and immune cell populations. Strikingly, chemotherapy was less effective in 3D vs. 2D at equivalent levels of uptake, highlighting the importance of retaining tumour microenvironment, extracellular matrix, and cell populations to better predict drug response in osteosarcoma.

## 2. Materials and Methods

### 2.1. Murine Osteosarcoma Cell Line and Culture

Murine osteosarcoma cell line (MosJ) was used for both in vitro and in vivo experiments. Cells were grown in DMEM high glucose (Lonza Group Ltd., Basel, Switzerland) supplemented with 10% FBS (Eurobio Scientific, Les Ulis, France) and 1% penicillin/streptomycin (Lonza). Depending on the number of cells required for in vitro or in vivo experiments, amplification was performed in either 75 or 175 cm^2^ tissue culture flasks (Corning, Wiesbaden, Germany). Cells were seeded at a density of 1 × 10^4^ cells/cm^2^ and incubated at 37 °C in a humidified tissue culture incubator with 5% CO_2_. After 3 or 4 days, when cells reached 70–80% confluence, they were detached using trypsin-EDTA (ethylene diamine tetra-acetic acid) (Eurobio), resuspended in DMEM supplemented with 10% FBS to inactivate enzyme activity, centrifuged at 500 *g* for 5 min and washed in PBS (Lonza). Cells were counted on a Malassez plate. MosJ cells were either seeded into 24 or 96 well plates for 2D cell culture experiments or injected into para tibial muscles of C57BL/6J mice to give rise to spontaneous osteosarcoma bone tumour growth as previously described [28].

### 2.2. Murine Models of Osteosarcoma

All in vivo experiments were performed with C57BL/6J mice housed in the animal facilities (Experimental Therapeutic Unit, Nantes, France) at the Faculty of Medicine of Nantes (Agreement D-44015). The murine osteosarcoma model was approved by the Regional Ethics Committee on Animal Experimentation (CEEA 6) and the Ministry of Research (APAFlS#8449-20170 1 0316591455 v3). For this study, four-week-old male mice from Janvier Labs (Le Genest-Saint-Isle, France) were used. After arrival, mice were left undisturbed in HEPA filtered and ventilated boxes containing 5 animals with pelleted food and drinking water ad libitum for at least one week before starting any experimentation. The room was air conditioned at 20 ± 1 °C with an artificial day/night cycle of 12 h. MosJ cells were amplified until 70–80% confluence in 175 cm^2^ flasks. The day of inoculation, cells were washed in PBS, trypsinized and rinsed in DMEM 10% FBS. Then, cells were washed in PBS to remove serum residues before cell counting and being resuspended in the appropriate volume of PBS. C57BL/6J mice were kept under general anesthesia using 4% isoflurane in air. The posterior right limb was disinfected with povidone iodine (betadine, Mylan Medical SAS, Paris, France) prior to cell injection. Intra-muscular paratibial injection of 2 × 10^6^ MosJ cells in 50 µL of PBS was performed using a syringe with a 29 G needle. Mice were observed for tumour growth twice per week until the tumour reached the approximate volume of 1000 mm^3^.

### 2.3. Vibratome Slicing of Osteosarcoma Tumours for Organotypic Culture

Mice were placed under general anesthesia as previously described before being euthanized by cervical dislocation. The para tibial tumour was carefully dissected and immediately placed in cold DMEM supplemented with 1% penicillin/streptomycin and stored at 4 °C for up to 2 h. Just before vibratome slicing, a small piece of tumour tissue was withdrawn using a scalpel in sterile conditions under a laminar flow cabinet (Optimal 18, ADS Laminaire, Aulnay-sous-Bois, France). The vibratome sections were taken following an axis perpendicular to the tibia bone, which allowed us to study the heterogeneity of the tumour. Fresh sections of the tumour were made using the vibrating blade microtome Leica VT1200S (Leica Biosystems SA, Nanterre, France) inside a vertical laminar flow hood. To ensure optimal viability of cells on the section surface, the Leica’s Vibrocheck measurement device was used prior to slicing. This system ensures minimal vertical deflection of the blade. For slicing, the tumour was placed in a tank containing cold DMEM. A cooling system maintained the sample at 4 °C during vibratome slicing. Serial 200 µm thick slices were performed using very low knife travel speed at a high vibration amplitude (2.5 mm/s). To culture organotypic tissue slices in vitro, basic medium was prepared with DMEM high glucose (Lonza) supplemented with 15% FBS and 1% penicillin/streptomycin (Eurobio). D-Glucose (Sigma Aldrich, Merck KGaA, Darmstadt, Germany) was added to medium at a final concentration of 6 g/L. Immediately after slicing, 200 µm thick sections were placed in a 6-well Ultra Low Attachment (ULA) plate (Corning) in 2 mL of culture medium. After 24 h of incubation at 37 °C with 5% CO_2_, the medium was discarded, and 6 mL of medium were added to keep slices in culture for 3 days. Then, the medium was refreshed every 3 or 4 days, with a minimal volume of 6 mL per well. Sections were kept in culture for up to 21 days, with analyses performed at days 4 and 8 to study the tumour microenvironment and drugs response.

### 2.4. Chemotherapy Drug Treatments

The effect of chemotherapy treatments was studied on both 2D and 3D organotypic explant culture. For 2D culture, MosJ cells were amplified, and 20,000 cells per well were seeded in 24 well plates in 1 mL of basal culture medium. After 24 h of incubation at 37 °C with 5% CO_2_, cells reached 30% confluence and drugs were added at different dilutions. For 3D organotypic culture, the same drugs were added 24 h after slicing to allow for stabilization of metabolic activity. Several drugs were tested: cisplatin (Santa Cruz Biotechnology Inc., Dallas, TX 75220, USA) and doxorubicin (Santa Cruz) were dissolved in sterile water and diluted in medium, while methotrexate (Santa Cruz), etoposide (Sigma Aldrich) and mafosfamide (Santa Cruz) were prepared in DMSO and diluted in medium from 0.01 to 1000 µM. For all drugs, 3 days of treatments were tested on 2D monolayer and 3D organotypic cultures, and the effect of chemotherapy was assessed at day 4 in both models as 2D cells reached confluence over this timeframe. For mafosfamide treatments, 96-well plates were used instead of 24-well plates, with 3160 cells per well. Drug responses were monitored by metabolic activity quantification (resazurin), histology, immunohistochemistry, multicolor flow cytometry and confocal microscopy. The dosages of methotrexate, etoposide, and mafosfamide for the tri-therapy treatment were calculated based on the French OS2006 study [29].

### 2.5. Cell Metabolic Activity Assay

Resazurin sodium salt (Sigma Aldrich) was used to measure metabolic activity on 2D and 3D organotypic culture. Resazurin stock solution was prepared at 0.5 mg/mL in sterile PBS. The working solution was prepared with 20% resazurin stock solution in basal or drug treated medium and added to the culture. After 3 h of incubation in the dark at 37 °C with 5% CO_2_, supernatants were collected, and fluorescence was measured at 600 nm in a dark 96-well plate with a fluorescence microplate reader (Berthold Technologies, Bad Wildbad, Germany). Measurements were recorded before and after applying treatments in order to track metabolic activity. For organotypic culture, each section was removed from the 6-well ULA plate and put into a 24-well ULA plate prior to the resazurin test. This allowed measurements in the same well size as 2D cultures and allowed elimination of cells that may have migrated out of the organotypic slice. For mafosfamide treatments, small organotypic sections were removed from 24-well ULA plates and placed into 96-well ULA plates just before the resazurin test.

### 2.6. Histology and Immunohistochemistry/Immunohistofluorescence

Organotypic cultures were fixed in 4% formaldehyde overnight and rinsed in PBS prior to dehydration and paraffin embedding. Samples were cut into 5 µm sections using a microtome (Leica). Sections were deparaffinized and stained with hematoxylin/eosin. For immunohistochemistry and immunohistofluorescence of Ki67 and cleaved caspase-3, antigen retrieval was performed on matching deparaffinized sections. Cleaved caspase-3 rabbit anti-mouse antibody (Cell signaling, 9664S), an apoptosis marker, and Ki67 rabbit anti-mouse (Abcam, Cambridge, UK), a marker for proliferation, were used separately and added to the slides overnight at 4 °C. For immunohistofluorescence, a secondary antibody goat anti-rabbit AlexaFluor 647 (Life Technologies, A21244) was added to the slices for one hour at room temperature. DAPI was added prior to mounting the slides with Prolong (Life Technologies, Fisher Scientific SAS, Illkirch, France). Acquisitions were performed with inverted fluorescence microscope Olympus IX73 (Olympus, Hamburg, Germany) and images were analysed with Fiji software. For immunohistochemistry after incubation with the primary antibody, a secondary anti-rabbit biotinylated conjugated antibody was used before adding streptavidin/peroxidase (Dako, Agilent Technologies Inc., Santa Clara, CA 95051, USA) and Gill-2 hematoxylin. Stained histology sections were digitalized by using an automated slide scanner (Nanozoomer, Hamamatsu, Japan) and images were analysed with Nanozoomer Digital Pathology software (NDP view 2, Hamamatsu, Hamamatsu City, Japan).

### 2.7. Single Cell Suspension Preparation and Flow Cytometry

Multicolour flow cytometry was used to study the tumour microenvironment associated with 3D organotypic cultures at days 0, 4 and 8 after vibratome slicing. For each tumour, 2 organotypic slices were dissociated and pooled to obtain at least 2 million cells. Dissociation was carried out with a gentle MACS dissociator (Miltenyi Biotec, Bergisch Gladbach, Germany) using an enzymatic mix made of DL liberase (Sigma Aldrich) at a final concentration of 0.2 mg/mL and DNase I (Sigma Aldrich) used at 40 µg/mL in a total volume of 2.5 mL of DMEM. After 45 min at 37 °C in the dissociation device, the cell suspension was collected, diluted in 10 mL of DMEM supplemented with 10% FBS and passed through a 40 µm cell strainer to remove doublets and aggregates. Cells were then centrifuged at 500 *g* for 5 min, resuspended in PBS and counted on a Malassez plate using Trypan blue exclusion. In order to avoid non-specific background, 1 million cells were incubated for 15 min at 4 °C with Seroblock FcR rat anti-mouse (Bio-Rad, Marnes-la-Coquette, France) in PBS-2 mM EDTA-0.5% BSA prior to staining by fluorescent conjugated antibodies. After 20 min in the dark at 4 °C, the cells were washed in PBS-2 mM EDTA and stained with Nir-Zombie (Biolegend, San Diego, CA 92121, USA) for 2 min in the dark at room temperature. After two washes in PBS-2 mM EDTA, cells were fixed with 4% paraformaldehyde for 20 min at room temperature. Then, cells were washed once in PBS-2 mM EDTA before being stored at 4 °C in the dark. Flow cytometry acquisitions were done between 1 and 5 days after fixation. To monitor the microenvironment on organotypic explant slices, two fluorescent conjugated antibody cocktails were set up. Antibodies were chosen according to the fluorochrome stain index, cell representation in the sample and antigen expression. Optimal concentration of each individual antibody was determined in a preliminary experiment to obtain the best staining with minimal background. Then, antibodies were used all together in a mix. For monocytes/macrophages, dendritic cells and lymphocytes, a 10-color mix was used: CD45 BV650 (BD Biosciences, Le Pont de Claix, France) diluted in PBS-2 mM EDTA-0.5% BSA at 1/2500, F4/80 BV421 (Biolegend) at 1/20, CD8 PercPCy5.5 (BD Biosciences) at 1/80, CD4 PE-CF594 (BD Biosciences) at 1/200, CD11c PE (BD Biosciences) at 1/125, CD11b AF700 (BD Biosciences) at 1/80, Ly6C PeCy7 (Biolegend) at 1/1000, Ly6G BUV395 (BD Biosciences) at 1/100, I-A/I-E AF488 (Biolegend) at 1/500 and NiR Zombie (Biolegend) at 1/5000 was used as viability marker [30,31]. For stromal cells, a 5-color mix was used: CD45 BV650 (BD Biosciences) at 1/2500, Ly6A/E FITC (BD Biosciences) at 1/100, TER 119 BV605 (BD Biosciences) at 1/100, CD140 BV711 (BD Biosciences) at 1/100 and Nir Zombie as a viability dye [32,33]. As doxorubicin has an autofluorescence property, another multicolor panel had to be set up to monitor the impact of doxorubicin on stromal cells: CD45 BV421 (BD biosciences) at 1/2000, Ly6A/E FITC (BD Biosciences) at 1/100, TER 119 APC (BD Biosciences) at 1/100, CD140 BUV395 (BD Biosciences) at 1/400 and Nir Zombie as a viability dye. Doxorubicin fluorescence was detected as BV605 on BD LSR Fortessa X-20 (violet laser excitation at 405 nm, emission filter 610/20BP). Compensations were set up using Versacomp Antibody Capture beads (Beckman coulter) and all acquisitions were done using BD LSR Fortessa X-20. All data were analysed with FlowLogic 7.2.1 Software (Inivai Technologies (Victoria, Australia), Miltenyi Biotec).

### 2.8. Immunofluorescence and Confocal Microscopy

Immunofluorescence was performed directly on organotypic sections in order to visualize the macrophages and stromal cells at days 0, 4 and 8 after vibratome slicing. Sections were fixed in 4% formaldehyde overnight and rinsed in PBS prior to immunostaining. For macrophages, F4/80 rat anti-mouse (Novus Biological, Littleton, CO, USA, NB 600-404) was used with Cleaved caspase-3 rabbit anti-mouse (Cell signaling, 9664S) to monitor apoptosis. Blocking buffer was prepared in PBS with 0.05% Triton (Sigma Aldrich), 1% BSA (Sigma Aldrich) and 1% Normal Goat Serum, added to the slides and incubated for 1 h at room temperature. Then, antibodies were added at optimal dilutions and incubated overnight at 4 °C. After washing 3 times in PBS at room temperature, secondary antibodies were added to the sections: goat anti-rabbit AlexaFluor 647 (Life Technologies, A21244, Fisher Scientific SAS, Illkirch, France) and goat anti-rat AlexaFluor 488 (Life Technologies, A11006, Fisher Scientific SAS) were incubated for 2 h at room temperature in the dark. Slides were washed 2 times in PBS and were incubated for 1 h with Phalloidin AlexaFluor 546 (Life Technologies, A22283, Fisher Scientific SAS) diluted in PBS. After two additional washes in PBS, DAPI (Life Technologies) was added and incubated for 15 min in the dark at room temperature. Organotypic culture sections were then mounted in Prolong (Life Technologies) between the slide and the cover slip. Slides were kept in the dark at room temperature for at least 48 h before imaging. For stromal cells imaging, Sca1 (Ly6-A/E) rat anti-mouse antibody (BD Biosciences, 557403) and cleaved caspase-3 rabbit anti-mouse (Cell signaling, 9664S) were used. As for F4/80 staining, the secondary antibodies goat anti-rabbit AlexaFluor 647 and goat anti-rat AlexaFluor 488, phalloidin AlexaFluor 546 and DAPI were used. All acquisitions were performed using confocal microscope A1 R Si (Nikon, Tokyo, Japan) and images were analysed with Fiji software.

### 2.9. Canine Osteosarcoma Organotypic Model

Dog osteosarcoma tumours were supplied by Oniris (Veterinary school of Nantes) and Vétocéane veterinary clinic after resection of tumours as part of veterinary care for the treatment of osteosarcoma. Osteosarcoma occurs spontaneously in dogs with an incidence 27 times higher than humans [34], making them an ideal candidate model to test the practicality of personalised ex vivo chemotherapy screening on organotypic cultures. As with the mouse model, dog biopsies were resected immediately, put in culture medium, and used for organotypic culture. Doxorubicin treatment was tested and monitored using the resazurin assay at day 4, and immunohistology at day 8.

### 2.10. Statistical Analysis

Measurements are reported as mean ± standard deviation (SD) or standard error of the mean (SEM) as indicated. Statistical analyses were performed using GraphPad Prism 8 (GraphPad Software, San Diego, CA, USA). Comparisons among groups were carried out with analysis of variance (one- or two-way ANOVA) when three cases were compared or with two-tailed t test when two cases were compared. Tukey’s, Dunnett’s, and Sidak’s test were used for post hoc analysis. Statistical significance was set at *p*-value < 0.05.

## 3. Results

### 3.1. Long Term Organotypic Slice Culture of Osteosarcoma

Schematic representation of the study design is shown in Figure 1a. Following injection of MosJ osteosarcoma cells into the posterior right limb of C57BL/6J mice, the animals were euthanized, and tumours were resected when the desired mass was achieved. Vibratome sectioning of resected tumours into 200 µm thick organotypic slices was performed. Organotypic tumour slices were cultured in vitro with/without chemotherapy treatment before being fixed and stained by H&E, Ki67 and cleaved caspase-3 to visualize tissue structure, proliferation and apoptosis as shown in Figure 1b. These images show that organotypic slices had sustained proliferation from day 1 to 4, with a marginal decrease in proliferation. Day 4 slices demonstrated a moderate increase in apoptosis relative to day 0. Cultured organotypic slices were also assessed for metabolic activity/viability over a 21-day period by Alamar Blue assay or dissociated and analyzed by flow cytometry using the NIR Zombie dead cell marker. The metabolic activity/viability at each day was normalised to day 1. At day 21, metabolic activity is approximately 60% using Alamar Blue and approximately 50% viability in the flow cytometry graph as presented in Figure 1c,d. Finally, as shown in Figure 1e, Ki67 staining of slices over 21 days shows sustained cell proliferation up to day 21, with little apoptosis induction as determined by cleaved caspase-3 staining. These results demonstrate the potential of organotypic culture for long term culture of osteosarcoma.

### 3.2. Organotypic Slice Culture Cell Populations

Flow cytometry analysis of dissociated organotypic slices at day 0, 4 and 8 identified the proportions of specific cell subpopulations in the osteosarcoma tumour. Gating strategies are presented in Appendix A.

Figure 2a shows CD45+ immune cells as a percentage of total cell population. At day 0, the CD45+ cell percentage is 30% of the total population but decreases significantly to ~10% at day 4 and ~5% at day 8 (*p* ≤ 0.01 and *p* ≤ 0.001, respectively), showing a depletion of total immune cells in the tissue model over time.

The proportion of macrophage, monocyte, and dendritic cell subsets within the CD45+ immune cell population are shown in Figure 2b, demonstrating that the Ly6CLow/medium macrophage cell sub-population represents 40–50% of the CD45+ population, and remains relatively stable from D0 to D8, increasing significantly at D4 (*p* ≤ 0.01). The Ly6CHigh monocyte population accounts for less than 2% of CD45+ cells at D0 and decreases to ~0.1% by D8. The CD11c+ dendritic cell subset comprises ~7% of the CD45+ immune population at D0, which decreases to less than 1% by D4 and D8.

Macrophage activation was assessed within the Ly6C+ sub-population by F4/80, MHCII labelling. Figure 2c shows that I-A/I-Elow F4/80+ non-activated macrophages (M0) comprise ~25% of the Ly6C+ population at D0, increasing to ~80% at D8 (*p* ≤ 0.001 and *p* ≤ 0.01, respectively) with a concurrent decrease in the percentage of I-A/I-E high F4/80+ activated macrophages (M1, M2) from ~75% at D0 to ~20% at D8 (*p* ≤ 0.01).

The Sca1+, CD140+ CAF population is shown in Figure 2d as a percentage of the total tumour cell population. At D0, CAFs make up 3% of total cells in the tumour, decreasing significantly to ~1% at D4 and D8, showing a depletion of CAFs in the tissue model over time, as seen in the immune cell population (*p* ≤ 0.01 and *p* ≤ 0.01, respectively).

In Figure 2e, confocal fluorescence imaging of organotypic slices labelled with F4/80 and Sca1 at day 0, 4 and 8 show decreasing monocyte/macrophage and CAF cell populations within the tumour over time, consistent with flow cytometry results.

### 3.3. Organotypic Slice Culture Chemotherapeutic Screening

The efficacy of mafosfamide, cisplatin, doxorubicin, methotrexate, and etoposide were assessed in organotypic slice culture at D4 as shown in Figure 3a–e. The drugs were tested at a range of doses and global viability was determined by flow cytometry. Proportions of viable immune cells (CD45+) and CAFs (Sca1+, CD140+) were measured relative to the total viable cells. The data is represented as viable proportion of each cell population following treatment compared to untreated controls.

For mafosfamide, the global viability remained unaffected until 100 µM, where complete repression occurred (*p* ≤ 0.001). The CD45+ population showed a similar response, with full repression at 100 µM (*p* ≤ 0.01). The CAF population demonstrated a distinct response, increasing significantly from ~100% to ~300% (*p* ≤ 0.01) in response to 1–10 µM mafosfamide before complete repression at 100 µM.

The global viability of 0.1–1 µM cisplatin treated organotypic slices remained at ~100%, decreasing significantly to ~50% in response to 10 µM and complete repression at 100 µM (*p* ≤ 0.01 and *p* ≤ 0.001, respectively). The viable CD45+ cells demonstrated a distinct pattern to the global population, increasing significantly to ~150% at 10 µM before complete repression at 100 µM (*p* ≤ 0.05 and *p* ≤ 0.01, respectively). Similarly, the viable CAF population demonstrated a significant increase to ~250% when treated with 1 µM cisplatin before decreasing significantly at 10 µM and complete repression at 100 µM (*p* ≤ 0.001, *p* ≤ 0.05 and *p* ≤ 0.05, respectively).

Methotrexate did not substantially repress global or CD45+ cells at any dose up to 1000 µM. The CAFs were the only sub-population to demonstrate a response to methotrexate, showing a non-significant increase to ~140% at 0.1 µM, before decreasing to ~50% at 100 µM. None of the cell populations investigated demonstrated greater than ~50% repression when treated with methotrexate.

Etoposide treatment of organotypic slices did not cause significant inhibition of global viability with only a slight decrease detected at the 100 µM dose. Inversely, we observed an overall increase in the proportion of CD45+ cells with increasing doses of etoposide, reaching its peak of ~175% at 10 µM (*p* ≤ 0.05). We also observed a ~60% increase in the proportion of viable CAFs in response to 1 µM etoposide, with a significant decrease to ~15% in 100 µM treated samples (*p* ≤ 0.01 and *p* ≤ 0.001, respectively).

Doxorubicin treatment demonstrated a similar effect to cisplatin, global viability remaining relatively unchanged with a moderate decrease at 10 µM before complete repression at 100 µM (*p* ≤ 0.001 and *p* ≤ 0.001, respectively). The CD45+ cells demonstrated a significantly higher proportion, up to 200% at 1 µM doxorubicin treatment, returning to ~100% at 10 µM and complete repression at 100 µM (*p* ≤ 0.01 and *p* ≤ 0.001, respectively). As with the CD45+ cells, the CAF population exhibited a substantial increase to ~175% in response to 0.1 µM doxorubicin, with a progressive decline to 75% and 0% at 1 µM and 10 µM respectively (*p* ≤ 0.05, *p* ≤ 0.01 and *p* ≤ 0.01, respectively).

Tri-therapy treatment of organotypic slices was performed with etoposide (CRD-33 µM), methotrexate (CRD-6.9 µM) and mafosfamide (CRD-3 µM) at 0.01×, 0.1×, 1× and 10× clinically relevant dose as shown in Figure 3f. This treatment did not cause a significant decrease in global viability over the range of doses. The proportion of CD45+ cells increased significantly to ~175% from 0.01× to 1× dose before decreasing to >50% at 10× dose (*p* ≤ 0.05 and *p* ≤ 0.05, respectively). The CAFs demonstrated a non-significant increase at 0.01× dose before significantly decreasing to >25% at 1× and 10× doses (*p* ≤ 0.05 and *p* ≤ 0.01, respectively).

Fluorescence imaging of untreated and 10 µM chemotherapy treated organotypic slices are shown in Figure 3g, with F4/80 and Sca1 showing the proportion of monocyte/macrophage and CAFs within the tumour following treatment.

### 3.4. Chemotherapeutic Screening in 2D vs. 3D

For the purpose of the screening response to chemotherapeutics, it was desirable to compare 2D in vitro and 3D organotypic culture for chemosensitivity in this study. The efficacy of multiple chemotherapeutics on cell metabolic activity in organotypic cultures was compared with 2D in vitro culture by the Alamar Blue assay, as shown in Figure 4a–e. For mafosfamide, there was a significant difference between the 2D and 3D at the 1 µM and 10 µM dosages, wherein we saw complete repression of 2D cell metabolism at 10 µM while 3D cultured cells remained at ~100% until complete repression at 100 µM (*p* ≤ 0.001 for both). Similarly, there was a significant difference between the 2D and 3D cisplatin treated samples at the 10 µM dose where metabolic activity is completely repressed in 2D while 3D remains ~100% before complete suppression at 100 µM (*p* ≤ 0.001). For methotrexate, complete inhibition of cell metabolism in 2D or 3D was not reached with doses up to 1000 µM. However, there were significant differences between the 2D and 3D at 10 µM, 100 µM and 1000 µM (*p* ≤ 0.001 for all). In etoposide treated samples, the 2D and 3D metabolic activity significantly diverged at the 10 µM and 100 µM dosages, where 2D metabolic activity was completely repressed at 100 µM while 3D was unchanged (*p* ≤ 0.001 for both). Doxorubicin treatment caused a significant divergence between 2D and 3D samples at the 1 µM and 10 µM dosages, decreasing 2D metabolic activity to ~25% and 0%, with 3D at ~100% and ~75%, respectively (*p* < 0.001 for both). Complete inhibition of 3D metabolic activity was achieved at 100 µM doxorubicin. In Figure 4f, tri-therapy treatment of organotypic slices was performed with etoposide, methotrexate and mafosfamide at 0.01×, 0.1×, 1× and 10× clinically relevant doses in 2D in vitro and in organotypic slices. Tri-therapy had no effect on cell metabolism in 3D across all doses tested, however, 2D samples showed a significant decrease at the 0.01× dose and decreased until complete repression of metabolic activity at 10× (*p* ≤0.05, *p* ≤ 0.001, *p* ≤ 0.001, *p* ≤ 0.001, respectively). These results highlight the contrast in sensitivity to chemotherapeutics between cells grown in commonly used 2D in vitro and 3D organotypic culture, emphasizing the importance of appropriate platforms for drug testing.

### 3.5. Doxorubicin Uptake and Viability in 2D vs. 3D

By utilising the fluorescence property of doxorubicin, we compared drug uptake with the relative cytotoxicity of doxorubicin in 2D and 3D organotypic slice cultures. In Figure 5a, Ki67 staining of organotypic slices following doxorubicin treatment for 3 days demonstrates sustained cell proliferation at 1 µM with an observable decrease at 10 µM and complete loss of proliferation at 100 µM. Cleaved caspase-3 staining exhibited no enhanced apoptosis up to 1 µM with a mild increase at 10 µM and complete apoptosis at 100 µM. In Figure 5b, representative dot plots from flow cytometry analysis show Zombie NIR staining and doxorubicin uptake in cells treated in 2D and 3D with increasing doses (0.1, 1, 10, and 100 µM). Uptake of doxorubicin occurred at a much lower dose in cells growing in 2D than in organotypic slice culture. In addition to taking up drug more readily when grown in 2D, the cells also exhibited a lower tolerance and greater cell death in response to doxorubicin. Graphed flow cytometry data for drug uptake vs. cell viability is shown in Figure 5c; this graph demonstrates the unequal uptake of drug between the two models. There was no uptake of doxorubicin at 1 µM in 3D, whereas in 2D, doxorubicin positive cells were ~75% at the same dose. At ~10 µM in 3D where doxorubicin positive cells were ~75%, viability remained relatively high at ~60%. This contrasts with cells grown in 2D, which exhibited viability of ~25% at the same level of doxorubicin positive cells. Figure 5d represents viability as a measure of drug uptake, demonstrating a significant divergence between 2D and 3D viability/drug uptake from 0.1–10 µM doxorubicin with a much closer correlation at 100 µM. These findings demonstrate the diminished chemosensitivity of cells in 3D compared to 2D, regardless of drug uptake.

### 3.6. Canine Osteosarcoma Organotypic Slice Culture

As a proof of concept for the application of 3D organotypic culture for personalised chemotherapeutic testing, canine osteosarcoma patient tumours were resected before vibratome slicing for organotypic culture and toxicity testing as shown in Figure 6a.

In Figure 6b, immunofluorescence images of a D1 organotypic slice with DAPI (blue) nuclear and phalloidin (red) F-actin staining show the 3D tumour structure and cell arrangement. Histological analysis of H&E stained untreated organotypic slices at D1, D4 and D8 demonstrated the progressive loss of non-adherent nucleated cells over time, predominantly from the intraosseous space, while cells in the osteoid are retained (Figure 6c). Alamar blue quantification of untreated D1 and D4 cell metabolic activity, presented in Figure 6d, showed no significant change over incubation time. In Figure 6e, Alamar blue analysis of D4 organotypic slices treated with increasing concentrations of doxorubicin showed 10 µM and 100 µM doses causing >50% decreased metabolic activity, demonstrating the effectiveness of this chemotherapeutic in the 3D canine tumour model.

Histological analysis of H&E stained doxorubicin treated organotypic slices at D8, shown in Figure 6f, demonstrated progressive loss of stromal cells with doses of 0.01–1 µM doxorubicin, and loss of the global cell population at 10–100 µM.

## 4. Discussion

Osteosarcoma has shown little improvement in clinical outcome over recent decades [3]. Many drug candidates that displayed promising results from in vitro screenings did not demonstrate any potency in vivo [35]. This low translational efficacy is largely a result of the limited ability of commonly used in vitro models to replicate the osteosarcoma microenvironment [36], demonstrating the need for novel preclinical screening strategies. There have been significant advances in the development of 3D in vitro models of osteosarcoma that strive to address many of the shortcomings of 2D culture, including tumour spheroids alone [20], spheroids cultured in biomimetic 3D matrices [21], and cell culture on 3D scaffolds [37]. However, no single preclinical model faithfully reproduces the full complexity of tumours in vivo [7,9].

Organotypic culture has previously been used for in vitro investigation of a range of tissues/cancers, including osteosarcoma [27,38,39], demonstrating its ability to preserve patients’ tumour/stromal cells, cytoarchitecture and both physiological and pathological settings. This organotypic culture method has also been used for screening cancer therapeutic agents [25,26,27], establishing its utility as a highly complex, reliable, cost-effective, and easy-to-handle method to study drug responses in vitro. In this study, an in-depth interrogation of the cell populations of a murine osteosarcoma organotypic culture was performed for the first time and the long-term viability and metabolic activity of osteosarcoma organotypic cultures were confirmed up to 21 days ex vivo. This study also represents the first application of organotypic culture as an ex vivo rapid platform for preclinical chemotherapeutic testing in osteosarcoma, confirming enhanced biomimetic ability over 2D in vitro screening. As proof of principle for the suitability of this platform for personalised chemotherapeutic screening in osteosarcoma, organotypic cultures from canine patients were tested in vitro for the first time.

Preliminary investigation of the tumour microenvironment of the human osteosarcoma cell line (K-HOS) in NMRI-nude mice (n = 1) was performed. Assessing proliferation by Ki67 staining of organotypic cultures demonstrated continued but diminishing growth over the 4-day incubation following vibratome slicing (Appendix A). Additionally, Ki67 and cleaved caspase-3 staining of D4 organotypic slices treated at scalar doses of cisplatin, mafosfamide, and methotrexate demonstrated proliferation and apoptosis induction in response to increasing doses of chemotherapeutics (Appendix A). Flow cytometry analysis of dissociated organotypic slices of these cultures at day 0 and 4 identified low monocyte and macrophage cell numbers in the tumour microenvironment at day 0, with almost complete loss of these populations at day 4 in culture (Appendix A). These findings demonstrated the necessity for incorporation of murine osteosarcoma in immune competent syngeneic C57BL/6J mice to facilitate the investigation of the osteosarcoma-immune interaction in vivo. For this reason, further investigation of human osteosarcoma in in vivo models was not pursued and studies were continued with murine osteosarcoma in syngeneic in vivo models.

Examination of the cell populations over the culture period confirmed the presence of native stromal and immune cell components within organotypic cultures. CAFs were found to constitute ~3% of the total cell population, which is consistent with previous reports of 2–10% within cultures derived from osteosarcoma patient biopsies [40]. The presence of this stromal cell population greatly enhances the physiological relevance of our model, as previous studies have outlined the recruitment and induced differentiation of MSCs to CAFs through Notch/akt signaling at the tumour site in osteosarcoma [8], and the metabolic reprogramming of osteosarcoma by CAFs acting as tumour-feeding cells [10]. CAFs have also been demonstrated to promote migration and invasion of osteosarcoma cells via microvesicle encapsulated miRNA [41], and to regulate neovascularisation, immune modulation, proliferation and chemoresistance in a paracrine manner [7,42], supporting the previous reports of the importance in targeting this cell population for enhanced chemotherapy efficacy [43]. Analysis of the mononuclear phagocyte cell population within organotypic culture identified macrophages accounting for ~40–50%, monocytes for ~1.5%, and dendritic cells for ~7% of the CD45+ immune cells closely resembling the proportions identified in a previous study of osteosarcoma patient tumours [12]. Numerous studies have previously implicated TAMs in osteosarcoma metastasis, with a higher density of M2-type TAMs in lung metastases compared to primary tumours, and enhanced invasive capability in osteosarcoma cells through TAM activation of the COX-2/STAT3 axis and epithelial to mesenchymal transition (EMT) [15]. Osteosarcoma cells drive the polarisation of TAMs to M2-phenotype through inhibition of Notch signaling, promoting tumour growth and metastasis [16]. Inhibition of M2-polarisation was correlated with decreased metastatic potential and stem cell-like properties in osteosarcoma [44]. TAMs have also been shown to promote angiogenesis and tumour progression through the activation of intracellular signaling pathways [13,19]. In addition to the TAM population, other immune cell subsets present in organotypic culture including dendritic cells and monocytes are reported to play significant roles in tumour progression strengthening the relevance of the organotypic model presented here and its suitability for drug screening [45,46,47]. Early cytotoxicity testing of organotypic cultures ex vivo is key to retain the most clinically representative setting, as the immune cell population decreases over incubation time. In addition to facilitating more clinically relevant testing of chemotherapeutic efficacy in osteosarcoma, the presence of these stromal/immune cells within organotypic culture enabled the investigation of cytotoxicity in these different cell populations. We report that many of the drugs tested not only failed to inhibit global viability in organotypic cultures but demonstrated enhanced viability of CAFs and CD45+ cells at low doses of chemotherapy compared to untreated cultures. This highlights the importance of accounting for these populations when screening chemotherapeutics to ensure clearance of tumour supportive and promoting cells rather than osteosarcoma cells alone.

In this study, we compared the metabolic activity of osteosarcoma cells in 2D culture and 3D organotypic cultures, revealing a significantly lower inhibitory effect of chemotherapy on cells grown in 3D. These findings are in agreement with a growing number of studies that reported enhanced chemoresistance in osteosarcoma cells when tested in 3D spheroids compared to 2D monolayers, via cell–cell and cell–ECM interactions within the heterogenous tumour, decreased drug penetrance and molecular mechanisms of resistance [20,21,22,23]. We report that certain treatments, particularly methotrexate and etoposide, were ineffective in organotypic culture. Importantly, we also found that doses up to 10× standard dose of tri-therapy comprised of methotrexate, etoposide and mafosfamide did not affect cell metabolic activity in our organotypic culture models in contrast to cells in 2D, which showed complete inhibition of cell metabolism at the 1× dose (Figure 4). The standard concoction of tri-therapy drugs used to treat osteosarcoma in France includes methotrexate, etoposide and ifosfamide, an analogue of mafosfamide [29,48], thereby representing a potential concern with this regimen.

Using the autofluorescence properties of doxorubicin we further investigated the divergent efficacy of chemotherapeutics between 2D and organotypic culture. Organotypic cultures exhibited significantly lower uptake of doxorubicin compared to 2D cells, demonstrating drug penetrance and uptake more representative of tumour cells in vivo. However, this was not the only reason why a lower drug effect was observed in 3D. Interestingly, we also identified substantially reduced sensitivity to doxorubicin treatment in 3D organotypic culture compared to 2D at equivalent levels of cellular uptake in both models (Figure 5). The tumour microenvironment in vivo comprises physical influences such as hypoxia, extracellular pH, ECM composition and vascular abnormalities [49], along with the different osteosarcoma interacting immune cells, CAFs, and their secretomes [50,51]. This diminished sensitivity in 3D culture is likely mediated by these TME elements in our model. The influence of cell effects is supported by studies that describe paracrine interactions between MSCs and osteosarcoma cells driving resistance to doxorubicin, whereby MSC-secreted IL-6 protects tumour cells by STAT3 activation from doxorubicin-induced apoptosis in vitro and in vivo [42]. Independent of interactions between different cellular subsets, arranging osteosarcoma cells in 3D orientation alone confers enhanced chemoresistance as shown by contemporaneous studies that reported diminished sensitivity to doxorubicin in osteosarcoma spheroids and 3D silk scaffolds compared to 2D monolayers [23,37,51,52]. Our study highlights one of the major advantages of organotypic culture, accounting for TME regulated mechanisms when screening potential therapies. The use of more complex models for early preclinical drug discovery and screening has immense potential to provide more physiologically relevant testing environments, improving the translation of drug design to treatment success in patients.

Personalised medicine represents a significant advance from standardised front-line therapies, enabling treatment stratification and prediction based on the characteristics of an individual patient’s tumour [53]. Osteosarcoma incidence is ~10 times more prevalent in dogs than in humans, thereby providing a significant patient cohort to test chemotherapeutic treatments [34,54]. The shorter canine lifespan and reduced survival times achieved in canine patients in response to combined surgery and cytotoxic chemotherapy allows for clinical trials in dogs to be conducted over a shorter time frame, facilitating more rigorous assessment of treatments before translation into human trials [55]. As proof of principle for ex vivo chemotherapeutic screening on patient-derived tumours, we tested doxorubicin efficacy on canine osteosarcoma tumours in organotypic culture for the first time. Cell metabolic activity was maintained in vitro over the testing period. However, from H&E staining, a visible loss of cells from the intraosseus space is observed, but cells within the osteoid persisted. This cell loss is comparable to the loss of immune cells observed in our murine organotypic cultures from day 1 to day 4 and is consistent with other studies of organotypic culture [56]. Doxorubicin treatment of canine patient organotypic culture demonstrated comparable response to treatment as observed in murine tumours, whereby a decreasing cell metabolic activity is observed with increasing concentrations of drug. It would be preferable to have larger sample sizes in future canine studies, with more cell populations probed. Nevertheless, together with the mice organotypic cultures, these findings further validate the utility of this model as a more cost-effective, biologically complex, and personalised platform for in vitro toxicity testing, reducing the demand on expensive and labour-intensive in vivo models for the selection of optimal chemotherapeutics. This innovative organotypic culture could be applied to the patient’s biopsy that is taken at the time of anatomopathology diagnosis of osteosarcoma allowing the selection of the most appropriate chemotherapy regimen within days. The robust nature of the organotypic culture method, verified by the ex vivo preservation of tumour–stromal interactions, native tissue morphology and cellular heterogeneity highlights its potential to better predict patient outcome and therapy response.

## Figures and Tables

**Figure 1 cancers-13-04890-f001:**
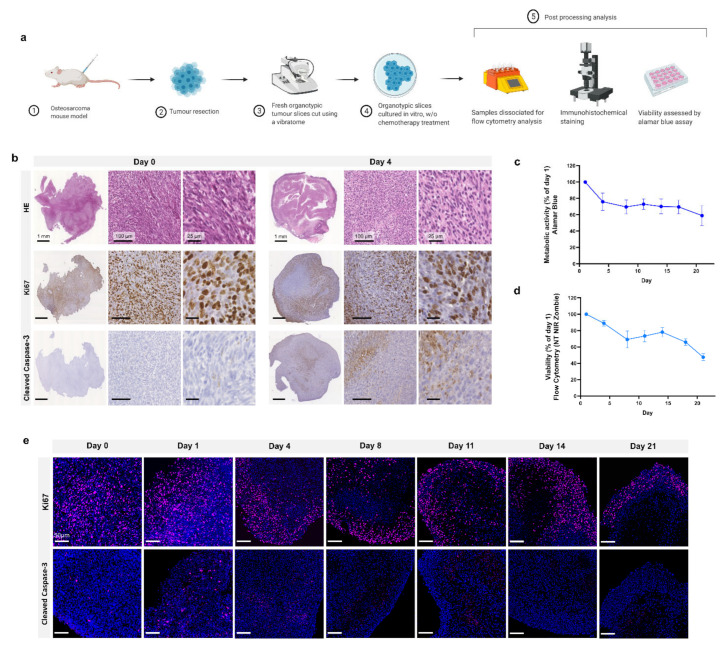
Organotypic culture. (**a**) Project workflow. Mice are injected with MosJ osteosarcoma cells; when desired mass is reached, mice are sacrificed, and the tumour is resected. Tumour slices are cut using a vibratome and cultured with/without chemotherapy treatments. Organotypic slices are fixed for immunohistochemical staining, dissociated for flow cytometry analysis, or assessed by Alamar blue assay. (**b**) 5 μm slices of fixed paraffin embedded organotypic sections from day 0 and day 4 are stained with H&E, Ki67, and cleaved caspase-3. (**c**) Metabolic activity and (**d**) cell viability of organotypic slices were measured for 21 days (Day 1, 4, 8, 11, 14, 17 and 21) and normalised to day 1; data is expressed as Mean ± SD of n = 6 samples and n = 3 samples, respectively. (**e**) Fluorescent microscopy on 5 μm slices of fixed paraffin embedded organotypic sections from day 0 to day 21. Samples are labelled with Ki67 (cell proliferation, pink) or cleaved caspase-3 (apoptosis, pink), and DAPI (nuclei, blue).

**Figure 2 cancers-13-04890-f002:**
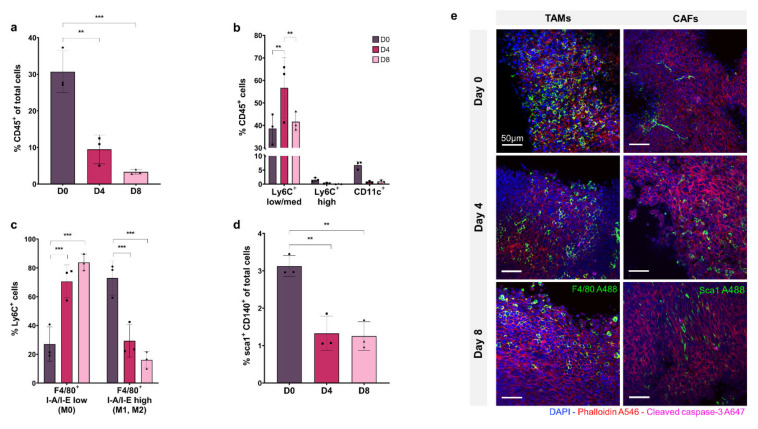
Flow cytometry analysis and fluorescent microscopy of organotypic slices cellular sub-populations at day 0, 4, and 8. (**a**) CD45+ immune cells as a % of total cells. (**b**) Breakdown of mononuclear phagocyte cells as a % of CD45+ population with Ly6C+ low/med macrophages, Ly6C+ high monocytes, and CD11c+ dendritic cells. (**c**) Breakdown of Ly6C+ F4/80+ I-A/I-E low non-activated macrophages and F4/80+ I-A/I-E high activated macrophages (M1, M2). (**d**) sca1+ CD140+ CAFs as a % of total cell population. (**e**) Organotypic slices labelled with DAPI (nuclei, blue), phalloidin (F-actin, red), cleaved caspase-3 (apoptosis, pink), and F4/80 (macrophage marker, green) or sca1 (CAF marker, green). Data is expressed as Mean ± SD of n = 3 samples. *p*-values; ** *p* ≤ 0.01, *** *p* ≤ 0.001.

**Figure 3 cancers-13-04890-f003:**
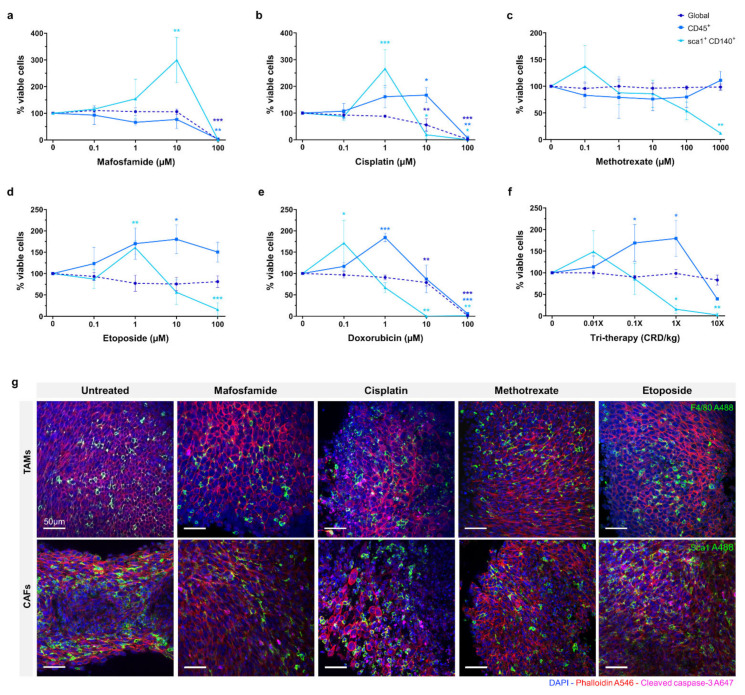
Effects of chemotherapy treatment on organotypic culture after 4 days. (**a**–**f**) Viability measured by flow cytometry assay after treatment with scalar doses of mafosfamide, cisplatin, methotrexate, etoposide, doxorubicin, and tri-therapy cocktail represented as % of untreated control. (**g**) Fluorescent images of 10 µM drug-treated slices at day 4. Samples are stained for DAPI (nuclei, blue), phalloidin (F-actin, red), cleaved caspase-3 (apoptosis, pink), and F4/80 (macrophage marker, green) or sca1 (CAFs marker, green). Data is expressed as Mean ± SD of n = 3 samples. *p*-values; * *p* ≤ 0.05, ** *p* ≤ 0.01, *** *p* ≤ 0.001.

**Figure 4 cancers-13-04890-f004:**
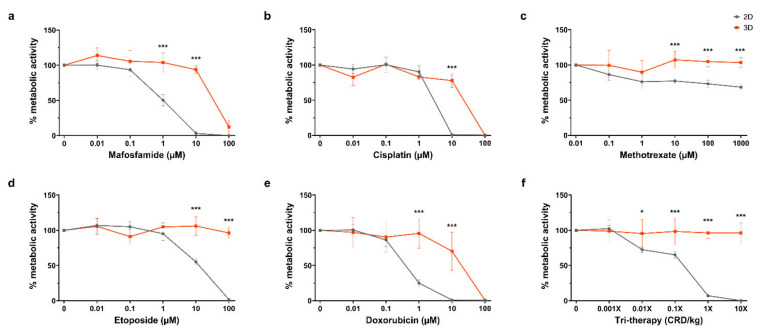
Drug response on 2D vs. 3D after 4 days. (**a**–**f**) Alamar blue assay after treatment with scalar doses of mafosfamide, cisplatin, methotrexate, etoposide, doxorubicin, and tri-therapy cocktail. Represented as % metabolic activity relative to untreated control D4/D1. Data is expressed as Mean ± SD of n = 4 samples. *p*-values; * *p* ≤ 0.05, *** *p* ≤ 0.001.

**Figure 5 cancers-13-04890-f005:**
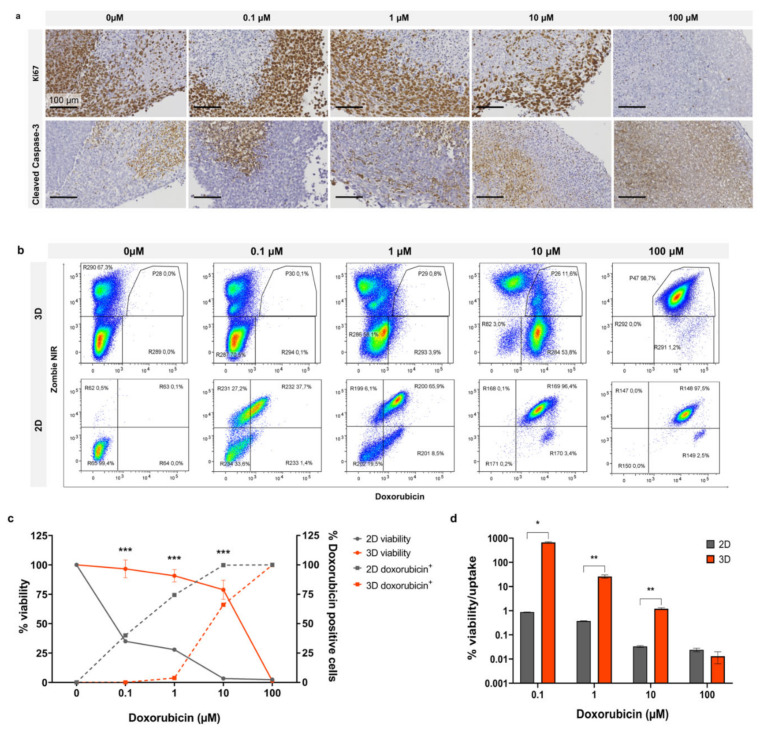
Doxorubicin effect in 2D vs. 3D after 4 days. (**a**) Organotypic slices treated with doxorubicin at scalar doses and stained for Ki67 or cleaved caspase-3. Flow cytometry (**b**), dot plots, and (**c**) line graph of Zombie NIR (viability) and doxorubicin uptake at scalar doses. (**d**) Cell viability as a function of doxorubicin uptake. Graphed data is expressed as Mean ± SD of n = 3 samples. *p*-values; * *p* ≤ 0.05, ** *p* ≤ 0.01, *** *p* ≤ 0.001.

**Figure 6 cancers-13-04890-f006:**
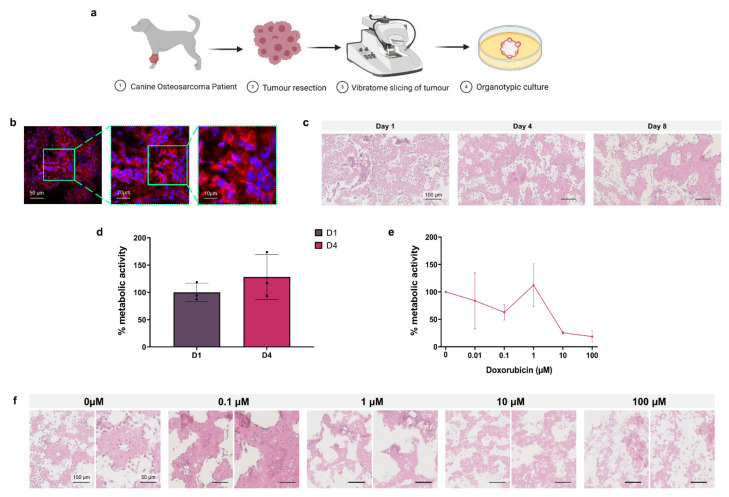
Osteosarcoma canine model. (**a**) Project workflow. Osteosarcoma is resected from the canine patient, organotypic slices are obtained by vibratome and are cultured in vitro. (**b**) Fluorescent microscopy of tumour slices labelled with DAPI (nuclei, blue) and phalloidin (F-actin, red). (**c**) H&E staining of organotypic slices at day 1, 4, and 8. (**d**) Metabolic activity of organotypic culture at day 1 and 4 expressed as % of day 1. Data is expressed as Mean ± SD of n = 3 samples. (**e**) Organotypic slices after treatment with scalar doses of doxorubicin at day 4 represented as % metabolic activity relative to untreated control D4/D1 of n = 3 samples. (**f**) H&E staining of organotypic slices treated with scalar doses of doxorubicin at day 8. Data is expressed as Mean ± SEM.

## Data Availability

The authors confirm that the data supporting the findings of this study are available within the article [and/or] its Appendix A. Raw data can be made available at the request of the authors.

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
