# Peer review of "Evaluation of the Chemotherapy Drug Response Using Organotypic Cultures of Osteosarcoma Tumours from Mice Models and Canine Patients"

_cancers, 2021, doi:10.3390/cancers13194890_

Round 1
Reviewer 1 Report
These is a nice and interesting study using murine and canine osteosarcoma organotypic models for Cytotoxicity testing of a range of chemotherapeutics drugs (mafosfamide, cisplatin, methotrexate, etoposide, and doxorubicin). A significantly decreased sensitivity was detected for chemotherapeutics in 3D organotypic culture compared to 2D monolayer. This study aims to recapitulate the 3D tumour multicellular microenvironment to better predict drug response.
The manuscript is well written, methods well described and results well presented.
Only few changes and comments are suggested:
Why drugs were tested a day 4? And in dog samples, in day 3 and 7?. I suggest to comment it in methods or results. Please, comment why only doxorubicin was assessed in dog samples.
Figure 1 (b): day 8 images could be included
In legend of figure 1 (c) Metabolic activity and (d) cell viability of organotypic slices: Please define the assayed days.
Figure 1 (e): it is not clear where this procedure is described in methods.
Finally, it should be interesting to compare all these results with normal bone cells and healthy tissue in terms of cytotoxicity. Perhaps, authors could comment it in discussion and limitations.
Author Response
Reviewer Comment: These is a nice and interesting study using murine and canine osteosarcoma organotypic models for Cytotoxicity testing of a range of chemotherapeutics drugs (mafosfamide, cisplatin, methotrexate, etoposide, and doxorubicin). A significantly decreased sensitivity was detected for chemotherapeutics in 3D organotypic culture compared to 2D monolayer. This study aims to recapitulate the 3D tumour multicellular microenvironment to better predict drug response.
The manuscript is well written, methods well described and results well presented.
Only few changes and comments are suggested:
Why were drugs tested a day 4? And in dog samples, in day 3 and 7?. I suggest commenting it in methods or results. Please, comment why only doxorubicin was assessed in dog samples.
Author Response: Thank you for the positive comments on our manuscript. Regarding the different days that the drugs were tested in mice and in dogs, in mice, all drugs were tested at day 4 only. This was because the aim was to compare 2D versus 3D, and it was not possible to maintain 2D culture beyond 4 days, due to cell confluence. The authors have added a sentence to methods section 2.4 to clarify this. Cytotoxicity was tested only in 3D in canine samples, as we did not have a canine osteosarcoma cell line to test in 2D. In order to be consistent with murine cultures, canine samples were assessed for metabolic activity of untreated and doxorubicin treated organotypic cultures at days 4 only. Doxorubicin alone was assessed in dog samples because this was the drug that we characterized most in mice due to its autofluorescence properties. Additionally, as canine biopsies were obtained from canine patients with spontaneously occurring osteosarcoma which were undergoing treatment, the availability of canine tumour slices was limited. We therefore focused on doxorubicin as our drug of interest to test the proof of concept for ex vivo chemotherapeutic testing of canine organotypic culture.
Reviewer Comment: Figure 1 (b): day 8 images could be included
Author Response: Immunohistochemical staining for Ki67 and cleaved caspase-3 was not performed at day 8 in murine organotypic sections, however, immunofluorescence imaging of Ki67 and cleaved caspase-3 was performed at day 8 and is shown in Figure 1 (e) which was extended up to day 21.
Reviewer Comment: In legend of figure 1 (c) Metabolic activity and (d) cell viability of organotypic slices: Please define the assayed days.
Author Response: Metabolic activity was assessed at days 1, 4, 8, 11, 14, 17 and 21. Cell viability was assessed by Flow cytometry at days 1, 4, 8, 11, 14, 18 and 21. These have since been detailed in the manuscript in Figure 1 legend.
Reviewer Comment: Figure 1 (e): it is not clear where this procedure is described in methods.
Author Response: For immunohistofluorescence, antigen retrieval was performed after deparaffinization. As for immunohistochemistry, Cleaved caspase-3 rabbit anti-mouse and Ki67 rabbit anti-mouse antibodies were used separately. After overnight incubation at 4°C with primary antibody, secondary antibody goat anti-rabbit AlexaFluor 647 (Life technologies) was added to the slices for one hour at room temperature. Then, DAPI was added prior to slides mounting with Prolong. Acquisitions were performed with inverted fluorescence microscope Olympus IX73 and images were analysed with Fiji software. This information has been added to the methods section 2.6.
Reviewer Comment: Finally, it would be interesting to compare all these results with normal bone cells and healthy tissue in terms of cytotoxicity. Perhaps, authors could comment it in discussion and limitations.
Author Response: The authors appreciate the reviewers' comments; however, we believe while interesting that this would be beyond the scope of the current study. The cytotoxicity of the drugs included in this study have been tested in various cell lines in other studies. Additionally, the focus of our study is the testing of different chemotherapeutics to assess efficacy in a more biologically relevant setting compared to 2D, not to compare the relative cytotoxicity in healthy bone cells relative to osteosarcoma cells. Finally, technical limitations prohibit us from testing these agents in healthy bone as vibratome is unable to section bone due to its mechanical properties in native form.
Reviewer 2 Report
The submitted paper is an interesting approach to investigate the impact of chemotherapeutic drugs in vitro. However, as organotypic tumor slices are used, that is probably as most as close to the in vivo situation as possible. As expected, major differences were found between 2D and 3D cultures concerning drug response. The presentation of the results is clear and I was impressed by the high quality of the histology images.
Concerning the number of immune cells, which are slightly enhanced after treatment with certain drugs (page 10, figure 3). I think as mentioned in the discussion, is sign of enhanced viability (or do the authors mean that they will proliferate in culture)?
There seems to be a peak in caspase 3 on day 1 of culture (page 7, figure 1), as afterwards positive cells are scarce. How do the authors interpret that? Might that be a result of adaptation to cell culture conditions?
Author Response
The submitted paper is an interesting approach to investigate the impact of chemotherapeutic drugs in vitro. However, as organotypic tumor slices are used, that is probably as most as close to the in vivo situation as possible. As expected, major differences were found between 2D and 3D cultures concerning drug response. The presentation of the results is clear, and I was impressed by the high quality of the histology images. Concerning the number of immune cells, which are slightly enhanced after treatment with certain drugs (page 10, figure 3). I think as mentioned in the discussion, is sign of enhanced viability (or do the authors mean that they will proliferate in culture)?
Author Response: Thank you for your positive feedback, particularly in relation to our histological images. Regarding the number of immune cells after treatment with certain drugs, we cannot say that it is a proliferation. Each cell population at day 4 (e.g. Global, CD45+, and sca1+) following treatment is normalized against the same cell population without treatment at day 4. Therefore, for each sub-population the untreated day 4 sub-populations represent 100% while the treated sub-populations are represented relative to the same population (e.g. D4 CD45+ treated/ D4 CD45+ untreated). Although the viability of several cell sub-populations is enhanced in response to treatment at specific chemotherapeutic concentrations, we cannot directly attribute this to proliferation as we are not measuring over time.
Reviewer Comment: There seems to be a peak in caspase 3 on day 1 of culture (page 7, figure 1), as afterwards positive cells are scarce. How do the authors interpret that? Might that be a result of adaptation to cell culture conditions?
Author Response: The authors are of the opinion that the observed peak at day 1 is as a result of the death of cells following the process of slicing and initiation of organotypic culture as cells unable to adapt to culture undergo apoptosis, following this peak, remaining cells exhibit a greater adaptation to organotypic culture, demonstrated by the minimal cleaved caspase 3 detected in subsequent days.
Reviewer 3 Report
In this manuscript the authors attempt to address the problem of investigating resistance to chemotherapeutic drugs in osteosarcoma by developing a novel 3D in vitro model isolated from mice following in vivo osteosarcoma injection.
The authors present a characterisation of the model based on viability and proliferation markers, followed by an analysis of immune cell sub-types within the cultured. Cell viability is quantified following treatment with chemotherapeutic drugs used clinically for the treatment of osteosarcoma. The data on differing responses between monolayer and 3D cultures is particularly interesting in the knowledge of corresponding clinical data on the effectiveness (or lack of) such chemotherapeutic agents upon osteosarcoma. The data on doxorubicin is particularly interesting as it suggests that despite uptake in 3D, cell viability endures more so than in monolayer. Finally, the authors compare this model with one based on resected tumours from canine osteosarcoma.
Overall, this is a novel, potentially significant and interesting manuscript. The data on drug responses in 3D compared to monolayer are especially worthwhile. However, there are outstanding issues relating to the design that require resolving.
Major comments:
In Figure 1, data are presented to show a ~40% reduction in metabolic activity and ~40% decrease in viability in the absence of any treatments. This suggests a serious weakness in the model employed. Do tumour spheroids of MOS-J cells created and maintained in vitro show any similar decrease?
Figure 1b shows images of histology, however it is not possible for the reader to know how representative these are or indeed if there is any significance. Both Ki67 and Caspase staining are readily quantifiable so there should be data on this included.
Figure 2 shows a depletion of immune cells. What is causing this depletion? Are these cells dying or migrating out of the slices or is it something else? Again, this is in the absence of any treatment with potential chemotherapeutic agents, so it suggests a further weakness in the model. Given the importance of immune cells in subsequent analysis with treatments, then surely an explanation for what is happening to these cells in the absence of treatment is also required.
In Figure 5, the Ki67 and caspase 3 staining should be quantified.
Figure 6 contains very little. 6b merely shows the cytoskeletal marker phalloidin and the nuclear marker DAPI. This is uninformative. The H+E stained sections clearly show more cells at day 1 than at subsequent time points. Did the cells die or migrate? If the model loses cells then how can it function as a model? The data in 6d only shows 2 data points per condition. The manuscript states n=3 so where is the third? In 6e, are the decreases in metabolic activity significant or is the experiment under powered? In the figure legend for Figure 6 it states that “P-values; * p ≤ 0.05, ** p ≤ 0.01, *** p ≤ 0.001” but there are no significant results shown. More importantly, the authors try to argue in the discussion that the canine model is comparable to the mouse model while lacking certain disadvantages (labour intensive, etc). However, on the basis of the data shown, the canine model is losing most of its cells between days 1 and 4 (H+E staining) and cell viability is only demonstrated with n=2. This is far from persuasive.
Apart from stating n=3 for Figure 6 (when actually n=2), the N values are missing throughout for figures 1c-d, 3a-f, 4a-f, 5c-d, and 6e.
Minor comments:
In section 2.1 should state that a Murine osteosarcoma cell line was used, not cell “lines” (singular, not plural)
The 4 and 2 in “1 x 104 cells/cm2” need to be superscript.
Section 2.2 should state “For this study, four-week-old male mice” without the additional ‘s’ on “weeks”
“ad libitum” should be in italics
In section 2.8, specific details of each antibody need to be provided; i.e. catalogue numbers or reference codes.
Author Response
In this manuscript the authors attempt to address the problem of investigating resistance to chemotherapeutic drugs in osteosarcoma by developing a novel 3D in vitro model isolated from mice following in vivo osteosarcoma injection.
The authors present a characterisation of the model based on viability and proliferation markers, followed by an analysis of immune cell sub-types within the cultured. Cell viability is quantified following treatment with chemotherapeutic drugs used clinically for the treatment of osteosarcoma. The data on differing responses between monolayer and 3D cultures is particularly interesting in the knowledge of corresponding clinical data on the effectiveness (or lack of) such chemotherapeutic agents upon osteosarcoma. The data on doxorubicin is particularly interesting as it suggests that despite uptake in 3D, cell viability endures more so than in monolayer. Finally, the authors compare this model with one based on resected tumours from canine osteosarcoma.
Overall, this is a novel, potentially significant and interesting manuscript. The data on drug responses in 3D compared to monolayer are especially worthwhile. However, there are outstanding issues relating to the design that require resolving.
Major comments:
Reviewer Comment: In Figure 1, data are presented to show a ~40% reduction in metabolic activity and ~40% decrease in viability in the absence of any treatments. This suggests a serious weakness in the model employed. Do tumour spheroids of MOS-J cells created and maintained in vitro show any similar decrease?
Author Response: The authors appreciate the reviewers' comments, however, while the organotypic model presents limitations, such as the outlined reduction in metabolic activity and viability over culture period, this model still recapitulates the complex tumour characteristics better than other models currently used for the intended purpose of ex vivo personalized chemotherapeutic screening. This application does not necessitate prolonged culture such as the 21-day time period over which the 40% decrease noted by the reviewer occurred. The analyses were therefore performed at Day 4 in order to maintain the heterogenous population of cells in the organotypic culture of the tumour biopsies. Additionally, recently researchers have shown that dynamic culture of organotypic models better supports the retention and viability of cells than standard culture. The authors therefore believe this model could be further optimized by using dynamic culture ex vivo. While tumour spheroids are very useful models for cytotoxicity testing the authors do not believe that direct comparison of the cell viability between these two models is appropriate as organotypic culture encompasses a substantially larger and more heterogeneous population of cells which would be expected to respond differently to spheroids developed in vitro using cell lines (1).
Reviewer Comment: Figure 1b shows images of histology, however it is not possible for the reader to know how representative these are or indeed if there is any significance. Both Ki67 and Caspase staining are readily quantifiable so there should be data on this included.
Author Response: The authors agree that quantification of Ki67 and caspase staining would add to the data presentation, however, this was not carried out as multiple fluorescent images of each sample were obtained, the images shown are representative of each sample and intended to give the reader a snapshot. Quantitative analysis of organotypic cultures was performed using more precise methods including flow cytometry which is more precise and Alamar blue which is more representative of the whole organotypic section.
Reviewer Comment: Figure 2 shows a depletion of immune cells. What is causing this depletion? Are these cells dying or migrating out of the slices or is it something else? Again, this is in the absence of any treatment with potential chemotherapeutic agents, so it suggests a further weakness in the model. Given the importance of immune cells in subsequent analysis with treatments, then surely an explanation for what is happening to these cells in the absence of treatment is also required.
Author Response: From figure 1 (c/d) we can see a decrease in overall metabolic activity and viability of approximately 30% over 8 days. In figure 2 we show the loss of immune cells CD45+ (which represent ~30% of the total cell population) indicating a loss of the immune cells from the model while retaining stromal cells. This is supported by Figure 2 (e) which also demonstrates the loss of immune cells over 8 days incubation. The authors are of the opinion that this decrease is largely as a result of migration out of the slice by immune cells which are largely suspension or weakly adherent cells, this opinion is supported by Figure 1 (e) which does not demonstrate any substantial expression of cleaved caspase 3 over 21 days.
Reviewer Comment: In Figure 5, the Ki67 and caspase 3 staining should be quantified.
Author Response: Instead of using histomorphometry, quantitative analysis of organotypic cultures was performed using more precise methods including flow cytometry and Alamar blue which are more accurate and representative of the whole organotypic section.
Reviewer Comment: Figure 6 contains very little. 6b merely shows the cytoskeletal marker phalloidin and the nuclear marker DAPI. This is uninformative. The H+E-stained sections clearly show more cells at day 1 than at subsequent time points. Did the cells die or migrate? If the model loses cells, then how can it function as a model? The data in 6d only shows 2 data points per condition. The manuscript states n=3 so where is the third? In 6e, are the decreases in metabolic activity significant or is the experiment under powered? In the figure legend for Figure 6 it states that “P-values; * p ≤ 0.05, ** p ≤ 0.01, *** p ≤ 0.001” but there are no significant results shown. More importantly, the authors try to argue in the discussion that the canine model is comparable to the mouse model while lacking certain disadvantages (labour intensive, etc). However, on the basis of the data shown, the canine model is losing most of its cells between days 1 and 4 (H+E staining) and cell viability is only demonstrated with n=2. This is far from persuasive.
Author Response: While the authors agree that Figure 6b provides limited information this image was included to give the reader an understanding of the 3D architecture and cell density of organotypic cultures. The authors apologize for the oversight on Figure 6d, this figure has been amended and uploaded for resubmission, whereby we excluded day 8 as we were only able to obtain 2 tumours for analysis for this time point due to limited availability of canine patient tumour tissue. Therefore, the new Figure 6d includes day 1 and 4 only with n=3 samples. As previously mentioned, the limited availability of canine patient tumours for this study and therefore low sample sizes may have resulted in the lack of significance observed. Note, these are dogs with spontaneously occurring tumours that were receiving veterinary treatment and therefore it would not be possible to have as many samples as those that were induced in our mouse in vivo model. We can see substantial deviation in the values obtained for 0.1 and 1µM Doxorubicin but without obtaining significant values. While the authors agree that there are limitations of the organotypic model, we do not agree that most of the cells are lost between days 1 and 4 (H&E staining). As for mice tumours, we may observe loss of non-adherent cells from the sections of canine tumours between day 1 and 4. Nevertheless, a vast majority of cells are retained over this time as evidenced by figure 6d (n=3). This is also consistent with the previously observed loss of a proportion of immune cells over this time period to migration and cell death as demonstrated in Figure 1e caspase 3 staining of the murine model.
Reviewer Comment: Apart from stating n=3 for Figure 6 (when actually n=2), the N values are missing throughout for figures 1c-d, 3a-f, 4a-f, 5c-d, and 6e.
Author Response: The authors apologize for this oversight and have added this information into the manuscript.
Minor comments:
Reviewer Comment: In section 2.1 should state that a Murine osteosarcoma cell line was used, not cell “lines” (singular, not plural)
The 4 and 2 in “1 x 104 cells/cm2” need to be superscript.
Section 2.2 should state “For this study, four-week-old male mice” without the additional ‘s’ on “weeks”
“ad libitum” should be in italics
In section 2.8, specific details of each antibody need to be provided, i.e. catalogue numbers or reference codes.
Author Response: These details have been corrected in the manuscript. The following references for antibodies have been added in the revised manuscript.
Cleaved caspase 3 rabbit anti mouse: Cell signaling 9664S
Sca1 rat anti mouse: BD biosciences 557403
F4/80 rat anti mouse: Novus biological NB 600-404
goat anti-rabbit AlexaFluor 647 (Life technologies) A21244
goat anti-rat AlexaFluor 488 (Life technologies) A11006
phalloidine AlexaFluor 546 (Life technologies) A22283
A new reference has been added:
Han, S.J., Kwon, S. & Kim, K.S. Challenges of applying multicellular tumor spheroids in preclinical phase. Cancer Cell Int 21, 152 (2021). https://doi.org/10.1186/s12935-021-01853-8
Round 2
Reviewer 3 Report
I thank the authors for adequately responding to most of my comments, however, there are still some outstanding concerns regarding Figure 6.
“While the authors agree that there are limitations of the organotypic model, we do not agree that most of the cells are lost between days 1 and 4 (H&E staining).”
This reviewer did not state that most of the cells are lost between days 1 and 4; instead; instead I wrote that the “H+E-stained sections clearly show more cells at day 1 than at subsequent time points”. I have counted the nuclei in the higher objective Day1 and Day 4 H+E images and find ~220 nuclei at Day 1 and ~160 nuclei at day 4, representing a ~27% decrease, substantiating this point. If the authors believe that this is due to outwards migration of immune cells between Days 1 and 4 then they should supply evidence and state clearly so in the text.
The data in Fig 6d is at odds with the representative H+E images shown. Is it not possible that cells that have migrated out from the organotypic slices are still in the well, and are therefore contributing to the metabolic activity without any longer being in the organotypic slice? This would explain how metabolic activity is maintained while cell numbers within the organotypic slice have decreased.
Two more minor points:
- the Y axis on Fig 6d is labelled “% metabolic activity”, but percentage of what? As neither column is at 100%.
- Day 8 removed from 6d figure, but manuscript and legend both still refer to day 8 in Fig 6d – needs amending
Author Response
Dear Editor
A point by point response to the Reviewer 3 is given.
The revised manuscript is attached.
Figures 5 and 6 are new ones.
Best regards,
Pierre Layrolle
